# *Vigna radiata* extracts in pumpkin and soya bean oil: A novel therapeutic approach for Alzheimer's disease

Haroon Amin[1], Shazia Anwer Bukhari[1]*, Zunera Chauhdary[2], Naheed Akhter[1], Maria Saleem[1]

1 Department of Biochemistry, Government College University Faisalabad, Faisalabad, Pakistan,
2 Department of Pharmacology, Government College University Faisalabad, Faisalabad, Pakistan

* Shaziabukhari@gcuf.edu.pk

## Abstract

*Vigna radiate* also known as mung beans, contains various bioactive compounds like polyphenols, flavonoids, and saponins. *V. radiata* therapeutic potential is enhanced by preparation of its extract in Pumpkin oil and soya bean oil by enrichment of bioactive compounds holding antioxidant, anti-inflammatory, and neuro-protective properties. The research study was aimed was to explore the healing endeavors of *V. radiate* pumpkin and soya bean oil extract in rectification of neuro-motor dysfunction and mental health decline in Alzheimer's disease (AD) rat model. After preliminary physico-phytochemical characterization and GC-MS analysis, AD model was established by administration of oral D-galactose and aluminum chloride 150 mg/kg each for 42 days daily. *V. radiate* extract in pumpkin and soya bean oil at doses 250 and 500 mg/kg was administered and rivastigmine (3 milligrams per kilogram) to treatment animals. To determine the cognitive decline and neuro-coordination dysfunctions behavioral tests were performed along with biochemical, neurochemical and histopathological analysis. ELISA and real time polymerase chain reaction were carried out to estimate the expression of tumor necrosis factor-α, Interleukine-6 and mRNA expression of neurodegenerative biomarkers. Gas chromatography Mass Spectrometry findings revealed the existence of favorable amount of neuro-defensive bioactive compounds in both oil extracts. *V. radiate* pumpkin and soya bean oil extract dose proportionally alleviated the behavioral dysfunctions, modulated the first line antioxidant enzymes and neurotransmitters s' level with anticholinesterase pursuits. The mRNA expression of AChE, IL-1β, TNF-α, IL-1α and β secretase were downregulated by these extracts treatment. *V. radiate* oil extracts also modulated the neuro-inflammatory protein expression and histopathological hallmarks in AD model animals. Therefore, it is purposed that *V. radiate* enriched extract in pumpkin and soya bean oil could be used to treat AD like memory dysfunction and motor symptoms.

## Introduction

Alzheimer's disease (AD) represents a subtle neurodegenerative condition and emerges as a prominent obstacle in contemporary public health during the 21[st] century. It stands as the primary contributor to dementia [1]. As the World's population gets older, the number of people

**Data availability statement:** All relevant data are within the manuscript.

**Funding:** The author(s) received no specific funding for this work.

**Competing interests:** The authors have declared that no competing interests exist.

affected by this serious condition is going up. It's expected to impact around 106.8 billion individuals by 2050. This emphasizes the crucial need to understand the detailed aspects of the condition and explore new and creative ways to research and intervene [2].

AD is identified by the genesis of neuritic plaques outside brain cells and accumulation of hyperphosphorylated tau proteins inside them, causing the death of neurons and loss of connections between them. This progression results in gradual cognitive decline [3]. The development of AD, a neurodegenerative condition, is associated with the presence of neurofibrillary tangles (NFTs) containing overly phosphorylated tau protein within cells and the accumulation of amyloid ß-plaques outside cells in specific regions of the human brain, particularly the cortical and limbic areas [3].

The proper functioning of most proteins relies on them adopting specific three-dimensional structures. Irregular protein folding has been connected to a growing number of health issues [4]. Some diseases result from mutations causing harmful changes in protein function. In these cases, certain proteins that are not stable can clump together, forming deposits of insoluble and toxic fibrillar proteins inside or outside cells [5]. Presently, approximately 30 human diseases are associated with protein misfolding and the formation of amyloid structures [6]. These diseases encompass neurodegenerative conditions like AD and Parkinson's disease (PD) [4].

Marked by a gradual deterioration in cognitive abilities, AD deprives individuals of their uniqueness, skills, societal contributions, and cherished memories. This imposes a substantial emotional and economic burden on families and societies, estimated to be approximately \$321 billion in the year 2022 [7].

Mung beans (*Vigna radiata* L.) have several polyphenols that can protect the brain, making them a potentially functional diet for preventing AD. A total of nineteen significant phenolic compounds were identified in green gram, comprising ten phenolic acids and nine flavonoids. By evaluating their amounts and optimal dosage in experiments using rodent models of AD, it is suggested that vitexin, isovitexin, sinapic acid, and ferulic acid may be the key potent compounds responsible for the neuroprotective effects of mung beans [8]. These plants have the potential to be an effective treatment for AD.

Green chemistry and green extraction principles aim to minimize environmental impact, reduce waste, and promote sustainability in chemical processes. By using vegetable oils for extraction, we align with these principles and contribute to a more environment friendly and sustainable future [9]. Therefore, the objective of the study was to prepare enrich *Vigna radiate* L. extract by using pumpkin and soya bean oil and to assess the neuro-protective effects of *Vigna radiata* L in an animal model of AD.

## Materials and methods

### Chemicals and drugs

Ethyl Alcohol, $CHCl_3$, Folin – Ciocalteau reagent (FCR), gallic acid ($C_6H_2(OH)_3CO_2H$), piperine, bovine serum albumin (BSA), NaOH, aluminum nitrite ($Al(NO_2)_3$), quercetin, aluminum chloride ($AlCl_3$), isopropanol, D-galactose, $H_2O$ for injection, and methyl Alcohol of laboratory grades were bought from sigmaAldrich (USA), rivastigmine from Novartis Pharma (Pak) Ltd; triazol (Invitrogen ™), cDNA kit (Thermo Scientific), cyber green (SYBR® Green master mix of Bio-Rad), primers (Thermo FisherScientific – US) were bought.

### Animals

Animals of either sex, mice (25-30 g) for oral acute toxicity study, rats (150-200 g) for neuro-protective or anti-Alzheimer's study were obtained from the animal house at Government

College University Faisalabad. They were sheltered in standard laboratory situations (25 ± 3ºC) with a twelve-hour bright and shady cycle and thirty to sixty percent dampness. The animals were allowed to move freely to meal in stainless steel separate cages. This animal study was approved and initiated after obtaining permission from the official review panel at GCUF.

## Institutional Review Board approval/Ethical statement

The Institutional Review Board (IRB) at GC University granted approval for animal trials, and GCUF/ERC/410 was the approved number. Every experimental protocol followed the guidelines set forth by the National Research Council's Institute of Laboratory Animal Resources, Commission on Life Sciences University, in 1996.

## Plant collection and extraction

Mung beans were collected from Arifwala, Pakistan, in July 2023 and authenticated by the Botany Department at the University of Agriculture Faisalabad (UAF), Pakistan. A voucher specimen was put forward to the herbarium (745-08-2023). The beans were washed to eliminate dirt and unnecessary constituents, and then dehydrated under dark for one month. They were ground to make pulverized stuff, and extracts were prepared in pumpkin oil and soybean oil using the green extraction technique and microwave-assisted extraction procedure.

## Green extraction by using pumpkin and soya bean oil

The plant material was weighed and added to 1000 mL beakers. The microwave oven was set to 9000 watts. In the first cycle, 900 mL of oil and 100 g of powder were combined in each beaker. The mixtures were heated for 2 minutes, followed by a 30-second pause. This procedure was carried out five times. Then, the supernatant was removed by means of filter paper. In the second cycle, the pellets were suspended with 500 mL of oil and the heating and filtration process was repeated five times. In the third cycle, 500 mL of oil was added to each beaker, and the process was repeated again. Finally, the filtrate was evaporated using a rotary evaporator at 40°C, yielding an oil-enriched extract used for further analysis [9,10].

## Physio and phytochemical characterization

The physicochemical properties were evaluated following the procedures outlined in the USP-National Formulary 2003. Various analyses were performed, including moisture content, total ash content, acid insoluble ash, water insoluble ash, sulphated ash, water-soluble extractive, alcohol-soluble extractive, total lipid content test, total protein content test, carbohydrate content test, total glycosaponins content test, estimation of secondary metabolites, total polyphenolic content test, total flavonoid content test, and total alkaloid test. These analyses encompassed a comprehensive assessment of the substance's chemical composition and properties [11].

## GC-MS analysis

The prepared extract was quantitatively analyzed by GCMS analysis to estimate the quantity and quality of neuroactive compounds isolated in the extract. An Agilent (7890B) gas chromatograph provided with an inert mass selective detector (5977B) and a DB-5MS GC column (30m length, 0.25mm internal diameter, 0.25μm film thickness) was utilized for this process. Inoculation of specimen (2uL) was performed in the split less mode, with adjustment of temperature of injector 250°C and temperature of interface 280°C. The oven temperature was programmed from starting value of hundred degrees Celcius for thirty seconds, stepped up to

340°C at 20°C/min for a minute. Helium (He) served as the carrier gas, and electron impact ionization was employed at –70 eV in full-scan mode. The overall running time span was thirty minutes. NIST Library 20.0 was utilized for the search report.

## Oral acute toxicity studies

Acute oral toxicity studies were performed according to OECD 425 guidelines, this study was performed on nulliparous non pregnant female mice to determine $LD_{50}$ and safe dose level of VRSO and VRPO, 2000 mg/kg dose of each treatment was given for 14 days and animals were observed for any toxic reaction.

## Neuro-protective studies

**Induction of Alzheimer's disease (AD).** AD was induced in experimental animals through oral administration of aluminum chloride and d-galactose at a dose of 150 mg/kg each for 28 days daily [11].

**Experimental design.** Animals were divided into 07 groups (n = 10, in each group): control (received distilled water), AD control (Alzheimer's disease (AD) like phenotype group, received aluminum chloride and d-galactose at 150 mg/kg each), standard group (received rivastigmine at 3 mg/kg), *Vigna radiata* L. pumpkin oil groups (VRP500 and VRP250) at doses of 500 mg/kg and 250 mg/kg respectively, and *Vigna radiata L.* soybean oil groups (VRS500 and VRS250) at doses of 500 mg/kg and 250 mg/kg respectively. This treatment was continued for 28 days. During the last week of experimental study animals were trained in Morris water maze for behavioral analysis. After completion of experimental treatment behavioral studies were performed to investigate the effect of VRSO and VRPO on AD associated motor and non-motor symptoms.

After sacrifice brain tissues were isolated principally hippocampus for biochemical analysis, neurotransmitter quantification and gene expression analysis. Blood samples were collected to isolate serum samples for quantification of ELISA test of pro-inflammatory cytokines.

**Anesthesia and euthanasia procedure.** Animals were sacrificed under anesthesia by inhalant anesthetic isoflurane 3-4% with 100% oxygen supply by precision vaporizer. Euthanasia procedure was performed according to AVMA guidelines [12]. Animals of weight greater than 200 g were euthanized by cervical dislocation and animals of weight greater than 200 g were sacrificed under inhalant anesthesia by decapitation by commercially available guillotines by trained personals.

**Neurobehavioral observations.** Morris water maze task, open field test, passive avoidance test, Y-maze test, hole board test, wire hanging test, elevated plus-maze task were performed according to reported protocols [13].

**Morris Water Maze test (MWM).** The Morris Water Maze (MWM) set up comprises of a spherical chamber occupied with milky water, with external cues for navigation. Training involves placing animals in the tank to find a hidden platform. Animals can employ praxis, taxis, or spatial strategies for navigation. Various opacifiers and tank sizes are used, and training protocols include hidden-platform acquisition, probe trials, and working memory testing. Video tracking systems are commonly used for quantification, measuring parameters like escape latency, path length, and swimming behavior. During last week of experimental study, animals were allowed to explore MWM tank without platform and it was considered as acclimation phase. After that for 04 consecutive days four trials/day from different quadrants were performed and animals were trained and guided to reach hidden platform in 01 minute. We considered that animal have learned the task if we find reduction in escape latency time (less than 20 seconds and path length less than 1 meter after 04 day training sessions. At 05 day probe trails were performed and escape latency period from each quadrant was recorded.

**Open field test.** The locomotor and habituation behaviors of twenty-one days treated animals were evaluated in a woody square case with measurement 40 x 60 x 50 cm. The surface of the field was separated into twelve equal squares. Rats underwent two consecutive turns, first for preparing and second for trial, each lasting 6 minutes. The number of crossings (movement across rectangles), rearings (standing on posterior limbs), and fecal matter were recorded in couple of turns. Crossings directed locomotors activity, whereas the decline in rearing between turns reflected a measure of habituation.

**Elevated plus maze test (EPM).** The maze assembly took place in an isolated room to eliminate any external disturbances, including noises, scents, or movement. Additionally, low-intensity white noise may have been introduced to the behavioral experiment room. It was crucial to remove any potential guidance cues, such as drawings on the walls, which could have biased the animals' activity within the maze. The experimenter ensured minimal noise and movement throughout the trial and refrained from using any strong-smelling products to prevent inducing anxiety in animals Illumination levels in the room were carefully controlled using a lux meter, with low-intensity lighting preferred for analyzing anxiogenic effects and higher intensity lighting for assessing anxiolytic effects. Following the adjustment of experimental conditions, the animals were given time to acclimate to the environment before testing begun. Prior to each test session, the maze was meticulously wiped with 70% ethyl alcohol to get rid of any potential contaminants. Once preparations were complete, the video camera was activated, and each animal was introduced into the maze for a 5-minute exploration period. The experimenter maintained a distance from the maze to avoid influencing the rat's behavior. After each trial, the maze was cleaned again to eliminate any lingering odors, and the next rat was tested in the same manner. This process continued until all animals had been tested. Recorded videos were later analyzed using automated software or manually with a chronometer, considering parameters such as entries in closed and open arms, time spent in each arm, and risk-assessment behavior.

**Y-maze test.** The evaluation of spatial memory was conducted using the Y-maze test, comprised of black plexiglass with three arms arranged at 120° angles to each other. During the learning phase, one arm of the maze was closed off, while various pictorial signs were strategically positioned nearby the setup. Rodents were then introduced into the experimental environment and permitted to freely travel the remaining two arms for duration of fifteen minutes. Following this exploration period, the rats were returned to their cages. After a 4-hour interval, the obstacle blocking one arm was removed, granting the rat's unrestricted access to all three arms for a 5-minute period. The performances of the rats were captured via camera and subsequently analyzed using Ethovision v1.90 software (Noldus). Parameters such as whole distance covered, average speed, incidence of appearing the new arm, and time devoted in the new arm were assessed from the recorded data.

**Passive avoidance task.** The short and long term memory deterioration in an associative manner was studied using a shock- motivated task. In order to prevent shock, test animals suffering from AD resist their innate bias. The set-up utilized in this experiment was made of Plexiglas measuring 27 cm by 27 cm by 27 cm, and it had grill work at the base made up of stainless steel rods spaced eight millimeters apart and with a thickness of three millimeters. Grid was connected to a battery to provide a 20V current source. A wooden rostrum measuring 10 cm by 7 cm by 1.7 cm was positioned at half height. The animals were placed on the elevated surface and dealt gently from the tip of their tails. In the initial trial, the animal is supposed to keep its position for fifteen to twenty- two seconds before stepping down to the floor. Two hours later, after the first trial the second trial begun. Rats were put up at wooden rostrum for test trial, and their step down time was observed.

**Y-Maze task.** Behavioral neurological science performs this test to assess rodents' short term memory, spatial memory and intellectual deficit. As opposed to the AD animal model, test animals with normal behavior prefer to voluntarily change the arms of the maze because animals with unaffected memory and prefrontal cortex functioning have a natural desire to find unique areas. The wooden apparatus used for this work has three arms joined at 120-degree angle in the shape of Y. these arms had a trigonal middle zone and measured 35 cm in length, 25 cm in height and 10 cm in width, for eight minutes, the animal's exploratory behavior in the Y maze was observed. To ascertain the spontaneous alteration, the rats were placed at the beginning point of each arm of apparatus, and their numbers of entries in each arm and number of triads (entry into three arms on consecutive selection) were recorded. Only when back paws fully penetrated to the arm, the arm entry was considered. The following formula was used to calculate spontaneous alteration:

$$Percentage\ spontaneously\ alterations = {triads\ completed} \Big/ {total\ entries\ into\ arms} \times 100$$

$$Laterality\ index = \frac{Movement\ towars\ left\ arm - movement\ towars\ right\ arm}{movemnet\ from\ left\ arm + movement\ toward\ right\ arm}$$

## Biochemical analysis

**Estimation of oxidative stress molecular markers.** *Formation of brain homogenate*: Isoflurane was administered as an anesthetic to each animal, and the animals were euthanized to remove their brains. Brains were cooled to -80 °C in a biomedical freezer after being cleaned with frozen NS (normal saline). After homogenizing the brain tissues in a tissue homogenizer, the supernatant was recovered by centrifuging the mixture for thirty minutes at four degrees Celsius at 800 rpm.

*Calculation of malondialdehyde (MDA)*: The degree of lipid peroxidation is revealed by the MDA level. To conduct this evaluation, 200 microliters of brain homogenate was mixed with fifteen percent trichloroacetic acid (TCA), 0.25 molar Hydrochloric acid, and 0.38% (w/w) thiobarbituric acid (TBA) in a falcon tube. The entire combination was maintained at 90 °C for 15 minutes in a hot tub prior to chilling. The solution was centrifuged for ten minutes at 4000 rpm after cooling [14]. The optical density at 532 nm was recorded after the top layer was collected. MDA level was estimated using this formula:

$$MDA\ level = \frac{Y \times vt \times 100}{E \times wt \times Vu}$$

Here Y represents absorbance, Vt is the test mixture's total volume (mL), E is the coefficient $1.56 \times 10^5$, wt is the brain's weight (grams), and Vu is the aliquot volume (mL).

*Measurement of Catalase Activity (CAT)*: The brain homogenate (50 μL), 1.95 mL of fifty milimolar phosphate buffer (pH 7.4), and 1 mL of thirty milimolar $H_2O_2$ were taken for this investigation. At 240 nm, the optical density was observed. CAT was calculated using the subsequent formula:

$$Catalase\ activity = \frac{Absorbance\ shift\ per\ minute}{Extinction\ co-effiecient \times sample\ volume \times mg\ of\ protein}$$

The extinguishment co-efficient, which is 0.071 mmol cm − 1, is determined by multiplying the sample volume (mL) by the protein milligrams in the brain homogenate.

*Estimation of superoxide dismutase (SOD)*: The task involved creating a 3 mL total combination by mixing 100 μL of tissue homogenate with point one molar potassium phosphate buffer at 7.4 pH (2.8 mL) and 0.1 mL pyragallol solution. Using a spectrophotometer absorbance was recorded at 325 nm and correlated with the SOD standard linear regression curve (unit/mL).

*Assessment of reduced glutathione (GSH) level*: In this experiment, 1 mL of tissue homogenate and the same concentration of trichloracetic acid were mingled together and centrifuged at 3000 rpm for 30 minutes. A mixture of point five milliliter DTNB and four milliliters of 0.1 molar phosphate buffer (7.4 pH) was added to the supernatant (2 mL). The OD of the specimen and control, which included all components but brain tissue homogenate, was measured at 412 nm. GSH amount was estimated by means of this formula:

$$GSH = \frac{absorbance - 0.00314}{0.0314 \times dilution\ factor \times aliquot\ voulme}$$

*Estimation of nitrite level*: The Griess mixture was prepared by mixing 2.5% $H_3PO_4$, point one percent N-1-naphthyl ethylene amine dihydrochloride, and one percent sulphanilamide. It was later equally combined with tissue homogenate to estimate the nitrite level. After 10 minutes of incubation, the test solution's absorbance at 546 nm was determined [14]. Following that, the OD at 546 nm was recorded.

**Determination of chemical transmitters of brain.** *Synthesis of aqueous part*: Combine the brain tissue homogenate with a 5 mL solution of HCl-n-butanol to prepare the aqueous phase. Centrifuge for 10 minutes at 2000 rpm. Centrifugation was used to extract the outer phase, which was then combined with two point five milliliters of heptane solution and point three milliliter of Hydrochloric acid and forcefully shaken. The resultant mixture was centrifuged once again for 10 minutes at 2000 rpm. Next, the organic and aqueous layers were separated to estimate the amount of neurotransmitters.

*Measurement of serotonin levels*: Aqueous phase (point two milliliter) was mixed to O-phthaldialdehyde and subjected to heat at 100°C for ten minutes for estimating serotonin levels. The resultant extract was allowed to cool to room temperature before having its absorbance at 440 nm. Conc. Hydrochloric acid (0.25 mL) was used as a blank.

*Determination of levels of noradrenaline and dopamine*: Aqueous phase 0.2 mL was obtained, and 0.1 mL of EDTA solution and 0.05 mL of HCL (0.4 M) were added to it. Next, 0.1 milliliter of $CH_3COOH$ and same amount of iodine in 0.1 mL of ethanol and $Na_2SO_3$ solution were added. The resulted combination was permitted to cool to ambient temperature (25 °C) after being preheated for six minutes at 100 °C. Next, absorbance for noradrenaline and dopamine was measured at 352 and 452. To prepare the blank for dopamine and noradrenaline, $Na_2SO_3$ was added before the iodine solution.

*Estimation of acetylcholinesterase (AChE) activity*: A tiny amount of brain tissue homogenate (0.4 mL) was combined with 2, 4 dithiobisnitrobenzoic acid (100 μL) and acetylthiocholine iodide (20 μL) in 0.1 molar phosphate buffer solution with a pH of 8.00 (2.6 mL). This test evaluated absorbance at 412 nm and generated a yellow tint when 2, 4 dithiobisniotrobenzoic acid and thiocholine reacted. In this experiment, the absorbance was taken at 412 nm and a yellow hue was formed by the interaction of 2, 4 dithiobisniotrobenzoic acid with thiocholine. The AChE activity was measured by following formula:

$$Acetylcholinesterse\ activity = \frac{5.74 \times 10 - 4 \times Absorbance\ shift\ per\ minute}{tissue\ concentration\ mg/mL}$$

*Histopathological analysis*: The brains were taken out and preserved in 4% formaldehyde. Paraffin-embedded brain tissues were cut into transverse sections using a scientific slicer (microtome) with a breadth of five micrometers, dyed with hematoxylin and eosin, and watched under a 10X compound microscope.

*PCR amplification in real-time*: The following primers were used in a PCR approach to identify the gene expressions linked to AD: α-synuclein, IL-1α, IL-1β, TNF-α, AChE (acetylcholinestrase), β-secretrase, and ABPP (β-amyloid precursor protein). Using a nanodrop spectrophotometer, the 260/280 nm absorbance ratio of RNA isolated using the TRIzol technique was measured. Afterwards, Thermo Scientific cDNA kit was used to transcribe RNA into cDNA. GADPH served as a house-keeping gene for the quantitative real time polymerase chain reaction (qRT- PCR) used to calculate the amount of gene expression. In the experiment, five microliters of complementary deoxyribonucleic acid (cDNA) and same amount of (five microliters) of cyber green were added to microplate wells along with a forward primer (0.5 μL) and corresponding reverse primer (0.5 μL). By inserting a microplate into the thermal cycler, 40 cycles of denaturation (breaking hydrogen of DNA stands) at ninety-five degrees Celcius, annealing (binding of primers to untwisted DNA strands) at sixty degrees Celcius, and extension (binding of DNA polyemerase to initiate elongation) at seventy two degrees Celcius were programmed. By inserting a microplate into the thermal cycler, forty cycles of denaturation at 95 °C, annealing at 60 °C, and extension at 72 °C were programmed (Fig 1).

*ELISA test of neuro-inflammatory markers*: The quantities of proinflammatory cytokines, such as TNF-α and IL-6, in serum were measured using an Enzyme Linked Immunosorbent Assay kit procedure created by Wuhan Zokeyo Biotechnology Co., Ltd., China (catalog number Y-83079-48T for TNF-α and Y-84561-48T for IL-6).

*Immune assay for SFRP4*: The SFRP4 expression in serum was determined using an ELISA. The mouse SFRP4 ELISA kit, developed by Wuhan Fine Biotech Com., Ltd., China, was used for this analysis [15,16]. The kit relied on sandwich enzyme-linked immunosorbent assay technology and was used for the quantitative detection of SFRP4 in serum.

| Biomarker | Sequence | BP size | Accession number |
|---|---|---|---|
| IL-1α | CCTCGTCCTAAGTCACTCGC | 102 | NM_017019.1 |
| | GGCTGGTTCCACTAGGCTTT | | |
| IL-1 β | GACTTCACCATGGAACCCGT | 104 | NM_031512.2 |
| | GGAGACTGCCCATTCTCGAC | | |
| Ach estrase | AGGACGAGGGCTCCTACTTT | 200 | NM_172009.1 |
| | CATGGCATCTCTCAGGTGGG | | |
| TNF α | GGAGGGAGAACAGCAACTCC | 168 | NM_012675.3 |
| | TCTGCCAGTTCCACATCTCG | | |
| β secretase | CCAACCTTCGTTTGCCCAAG | 197 | NM_019204.2 |
| | GCGGAAGGACTGATTGGTGA | | |
| Amyloid β precursor protein | GAGGTAGTCCGAGTTCCCAC | 127 | XM_006248012.3 |
| | GCTTGGCTTCCAACCTCTCT | | |
| SFRP4 | AGGCAATAGTCACTGACCTTCC | 129 | XM_063276786.1 |
| | CCTTTTTGCACTTGCACCGAT | | |
| GADPH | GGAGTCCCCATCCCAACTCA | 173 | XM_017592435.1 |
| | GCCCATAACCCCCACAACAC | | |

**Fig 1. Primer size and sequence information for RT-PCR analysis.**

## Statistical analysis

The data was displayed as mean ± SEM. Graphpad Prism version 6 was used to do One Way and Two-way ANOVA and Tukey's post-hoc test.

# Results

## Characterization through preliminary physico and phytochemical analysis

Physico and phytochemical analysis revealed that all parameters of physicochemical analysis were in acceptable range according to USP and national formulary standard limits. As presented in Table 1. Results of primary and secondary metabolites revealed the existence of favorable concentration of bioactive compounds (Tables 1 and 2).

## GC-MS analysis

GC/MS examination disclosed the potentiality of multiple bioactive compounds in VRSO extract and VRPO extract. As presented in Table 3. Linoleic acid higher concentration is present in VRSO similarly, VRPO showed the maximum concentration of oleic acid. Both compounds exhibited the robust neuroprotective potential as reported in previous studies (Figs 2 and 3).

## Acute oral toxicity study

After administration of limit dose 2000 mg/kg of VRSO and VRPO, no change in behavior was observed with any morbidity and mortality. Hematological, biochemical, liver function test and kidney function test revealed that VRSO and VRPO showed no toxic result on normal function of vital organs (Tables 4–6).

**Table 1. Physicochemical characterization of *V. radiate* bean powder.**

| Parameters | %age |
|---|---|
| Humidity contents | 5.25% |
| Total ash contents | 4.5% |
| Acid insoulble ash | 0.5% |
| Water insoluble ash | 1.5% |
| Sulphated ash | 18% |
| Alcohol soluble extractives | 13.73% |
| Total lipids contents | 1.9% |
| Total proteins | 0.87% |
| Total carbohydraes | 87.48% |

**Table 2. Phytochemical characterization of *V. radiate* pumpkin and soya bean oil extract.**

| Phyto-constituents | VRPO extract (%) | VRSO extract (%) |
|---|---|---|
| Total Proteins | 18.6 ± 0.1 | 10.8 ± 0.06 |
| Total Glycosaponins | 19.4 ± 0.2 | 31.1 ± 0.006 |
| Total polyphenols | 4.5 ± O.04 | 5.42 ± 0.2 |
| Total flavonoids | 51.4 ± 1.0 | 44.39 ± 0.2 |
| Total alkaloids | 15.2 ± 0.03 | 16.8 ± 0.03 |

**Table 3. Potent active compounds discovered in GC-MS examination through NIST library.**

| No. | VRSO | | | VRPO | | |
|---|---|---|---|---|---|---|
| | Retention time | Compound name | Neuroplasticity | Retention time | Compound name | Neuroplasticity |
| | 15.011 | Linoleic acid | Promoted synaptic plasticity, memory and cognitive functions [17] | 12.866 | Oleic acid | Neurotrophic factor, involved in dendrite and axonal development, promote neuronal migration and synapses [18] |
| | 19.561 | 9-Octadecenoic acid | Decreased neuro inflammation and improve cognition [19] | 15.603 | Linoelaidic acid | Promote neuronal plasticity and cognition [20] |
| | 22.062 | 4-Phenylpyridine | Unknown neuroprotective potential | 19.567 | Octadecadienoic acid | Modulate the expression of BDNF(brain derived neurotrophic factor), increase synaptic plasticity, decreased neuroinflammation and oxidative stress [21] |
| | 24.67 | gamma.-Tocopherol | Improve neurobehavioral cognitive functions by regulation of lipid signaling pathways, stress adaptation and regulation of protein c kinase activity [22,23] | 22.950 | Squalene | Promote myelination, prevented dopamine depletion, and locomotor impairment [24] |
| | 25.210 | Vitamin E | Neuro-defensive in Parkinson's disease model [25] | 25.578 | Sesamin | Neuroprotective in cerebral ischemia [26] |
| | 26.00 | Stigmasterol | Antioxidant and neuroprotective effect [27] | 29.349 | Chrysin | Promote neurogenesis and modulate neurotransmitters level |
| | 26.527 | Gamma-Sitosterol | Antidepressant effects [28] | 26.587 | Chondrillasterol | Neuroprotective and Antioxidant [29] |
| | 29.198 | Chrysin | Promote neurogenesis and modulate neurotransmitters level [30] | | | |

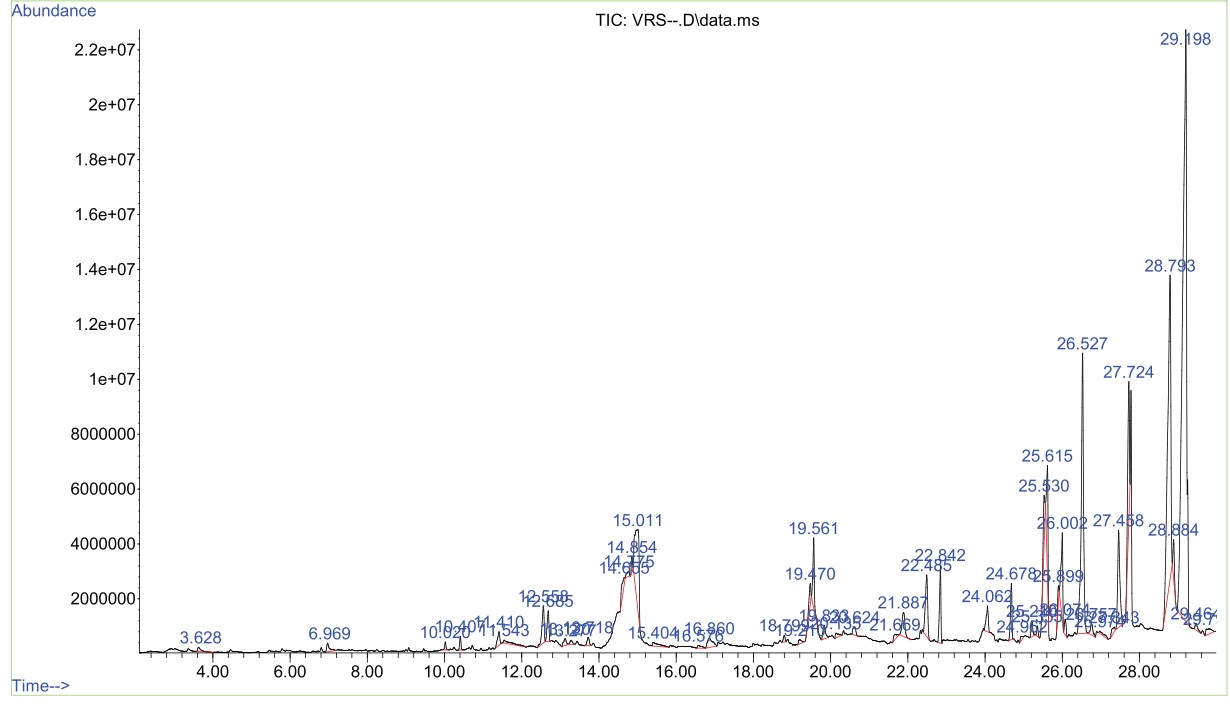

**Fig 2. GC-MS spectrum of detected bioactive compounds in VRSO extract.**

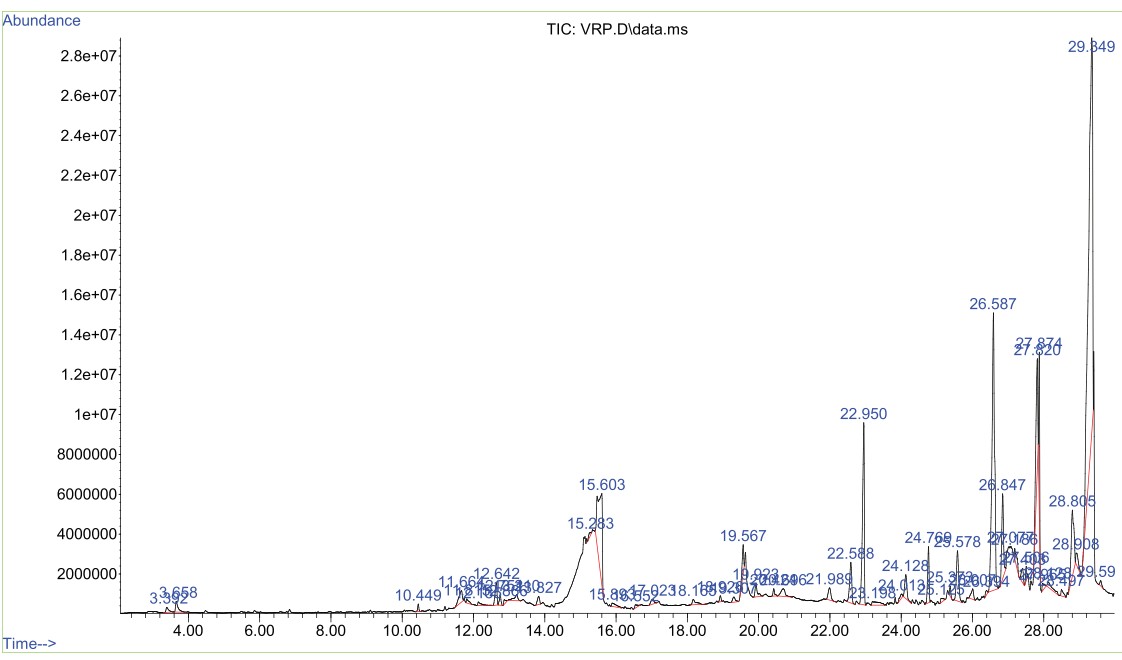

**Fig 3. GC-MS spectrum of detected bioactive compounds in VRPO extract.**

**Table 4. Hematological and biochemical analysis after acute oral toxicity study.**

| Parameters | Units | Normal Control group | Treatment group VRPO | Treatment group VRSO |
|---|---|---|---|---|
| Hemoglobin (Hb) | g/Dl | $12.53 \pm 0.1$ | $13.4 \pm 0.1^{ns}$ | $12.24 \pm 0.11^{ns}$ |
| Red blood cells (RBCs) | $\times$ 10.e6/ul | $8.3 \pm 0.1$ | $8.9 \pm 0.2^{ns}$ | $9.0 \pm 0.3^{ns}$ |
| Hematocrit (HCT) | % | $45.4 \pm 0.4$ | $47.2 \pm 0.3^{ns}$ | $46.1 \pm 0.1^{ns}$ |
| Mean corpuscular volume (MCV) | fl | $50.4 \pm 0.2$ | $53.2 \pm 0.2^{ns}$ | $51.0 \pm 0.01^{ns}$ |
| Mean corpuscular hemoglobin concentration (MCHC) | Pg | $14.1 \pm 0.2$ | $14.9 \pm 0.2^{ns}$ | $14.2 \pm 0.01^{ns}$ |
| Red cell distribution width coefficient of variation (RDW-CV) | % | $29.3 \pm 0.4$ | $28.7 \pm 0.1^{ns}$ | $29.2 \pm 0.21^{ns}$ |
| Total leukocytes count (TLC) | % | $9.4 \pm 0.3$ | $11.1 \pm 0.08^{ns}$ | $10.2 \pm 0.01^{ns}$ |
| Neutrophils | $\times$ 10^3/uL | $14.0 \pm 0.6$ | $17.9 \pm 0.3^{ns}$ | $16.6 \pm 0.13^{ns}$ |
| Lymphocytes | % | $61.9 \pm 0.2$ | $64.2 \pm 0.08^{ns}$ | $63.2 \pm 0.11^{ns}$ |
| Monocytes | % | $8.8 \pm 0.1$ | $7.8 \pm 0.2^{ns}$ | $8.2 \pm 0.12^{ns}$ |
| Platelet count | $\times$ 10^3/uL | $365.3 \pm 3.18$ | $371.3 \pm 2.40^{**}$ | $370.3 \pm 1.4^{ns}$ |

Statistical assessment (n = 10, in each group), One-Way ANOVA showed non-significant variations compared to control group.

So that both extract have LD$_{50}$ value greater than 2000 mg/kg. Therefore we selected dose level of both extract for further neuro-protective study in the light of LD$_{50}$ value. Histo-pathological analysis also revealed the intact architecture of vital organ tissues of VRSO and VRPO treated animals similar to control group animals (Fig 4).

**Table 5. Effect of VRPO and VRSO on renal Function Test (RFT).**

| Parameter | Unit | Control group | Treatment group VRPO | Treatment group VRSO |
|---|---|---|---|---|
| Urea | mg/dl | $45.4 \pm 0.2$ | $41.7 \pm 0.9^{ns}$ | $43.1 \pm 0.1^{ns}$ |
| Creatinine | mg/dl | $0.6 \pm 0.006$ | $0.5 \pm 0.006^{ns}$ | $0.55 \pm 0.02^{ns}$ |

Statistical assessment (n = 10, in each group), One-Way ANOVA showed non-significant variations compared to control group.

**Table 6. Effect of VRPO and VRSO on liver Function Test (LFT).**

| Parameter | Units | Control group | Treatment group VRPO | Treatment group VRSO |
|---|---|---|---|---|
| Total Bilirubin | milligram/deciliter | $0.7 \pm 0.01$ | $0.7 \pm 0.03^{ns}$ | $0.72 \pm 0.13^{ns}$ |
| Direct Bilirubin | milligram/deciliter | $0.3 \pm 0.006$ | $0.4 \pm 0.03^{ns}$ | $0.38 \pm 0.02^{ns}$ |
| Indirect Bilirubin | milligram/deciliter | $0.2 \pm 0.006$ | $0.33 \pm 0.07^{ns}$ | $0.38 \pm 0.06^{ns}$ |
| SGPT | mg/dl | $32.3 \pm 0.02$ | $29.8 \pm 0.02^{ns}$ | $30.18 \pm 0.12^{ns}$ |
| SGOT | mg/dl | $29.2 \pm 0.01$ | $31.0 \pm 0.3^{ns}$ | $32.01 \pm 0.11^{ns}$ |
| Alkaline phosphatase | IU/L | $120 \pm 3.0$ | $124 \pm 2.6^{ns}$ | $122 \pm 0.05^{ns}$ |
| Cholesterol | mg/dl | $157.6 \pm 0.8$ | $162.0 \pm 0.6^{ns}$ | $158.0 \pm 0.1^{ns}$ |
| Triglycerides | mg/dl | $126.6 \pm 0.8$ | $112.0 \pm 1.0^{ns}$ | $120.0 \pm 0.01^{ns}$ |
| High density Lipoproteins (HDL) | mg/dl | $48.7 \pm 0.7$ | $53.3 \pm 0.9^{ns}$ | $50.13 \pm 0.4^{ns}$ |
| Low density Lipoproteins (LDL) | mg/dl | $110.0 \pm 0.6$ | $88.3 \pm 0.8^{ns}$ | $90.3 \pm 0.02^{ns}$ |

Statistical assessment (n = 10, in each group), One-Way ANOVA showed non-significant variations compared to control group.

## Behavioral studies

**Morris water maze task (MWM).** Findings of Morris water maze task revealed that in AD like phenotype group escape latency (an earliest measure of spatial learning and memory is a time to reach hidden platform) was increased (p < 0.001) significantly compared to normative treatments and standard care group. The average velocity was markedly decreased and the distance travelled was also expanded in AD like phenotype compared to other groups due to thigmotaxis behavior of animals. However, treatment with VRSO and VRPO dose dependently improved learning and spatial memory capability in treatment groups in Morris water maze tank (Fig 5).

**Open field task.** It is revealed from open field task that animals in all groups reside longer at edges of apparatus contrast to expedition time at center of apparatus. In AD like phenotype group, the frequency and duration of rearing behavior of animals significantly declined (*p < 0.001*) compared to normative, standard care and treatment groups. In AD like phenotype group, stretch attends posture, freezing moments and defecation observed more frequently (*p < 0.001*) relative to other treatment groups. In VRPO and VRSO treatment groups animals travelled the longer distance at periphery of chamber, crossed the greater number of lines with greater velocity (*p < 0.001*) dose dependently compared to AD like phenotype group. The grooming behavior of animals indicated the self- grooming, maintenance of fur and wounds, exploration and relaxation attitude by licking, nibbling, scratching and cleaning. The grooming social behavior significantly declined (*p < 0.001*) in AD like phenotype compared to normative and treatment groups (Table 7).

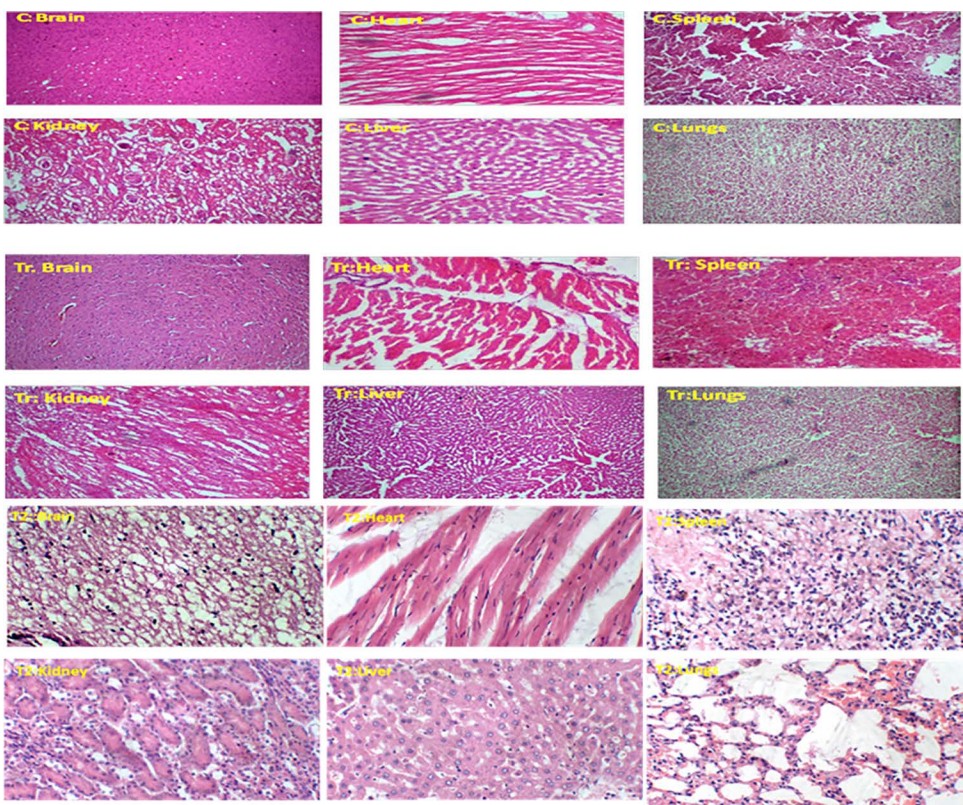

**Fig 4. Oral acute toxicity study: Effect of VRPO and VRSO (2000 mg/kg) on vital organs histopathology, C; control, Tr; VRPO, T2: VRSO.**

**Elevated plus maze task.** Rats' transfer latency (TL), a measure of the acquisition and retention of memory on an elevated plus-maze, was used to examine how VRPO and VRSO affected the learning and memory. It was observed that transfer latency was remarkably boost ($p < 0.001$) in AD like phenotype category contrast to normative category due to acquisition deficit induced by D-galactose and aluminum chloride. Although, curing by VRPO 250 & 500 mg/kg and VRSO 250 & 500 mg/kg increased consolidation, retrieval, acquisition and retention as manifested by significantly ($p < 0.001$) decline in transfer latency in treatment groups compared to AD like phenotype animals (Fig 6).

**Passive avoidance task.** This task was designed to investigate fear conditioning and inhibitory avoidance in AD like phenotype group in which D-galactose and aluminum chloride administration impaired the aversive learning and fear conditioning compared to normative group. It was manifested that the step down response time was minimized in AD like phenotype category ($p < 0.05$) contrast to dose receiving category. VRSO and VRPO treatment groups showed the intact ability to learn and avoid an aversive stimulus. The step down latency was significantly increased in VRSO and VRPO treatment groups (Fig 7).

**Effect of VRPO and VRSO on Y-maze task.** In Y-maze task total arm visits, spatial memory, score in triadic response, autonomic alternation score (%) and hemispheric dominance index were notably ($p < 0.001$) decreased in AD like phenotype category compared to normative, standard care, VRPO and VRSO treatment groups. However, treatment with VRSO and VRPO remarkably recovered spatial memory, learning and cognitive flexibility in Y-maze task (Table 8).

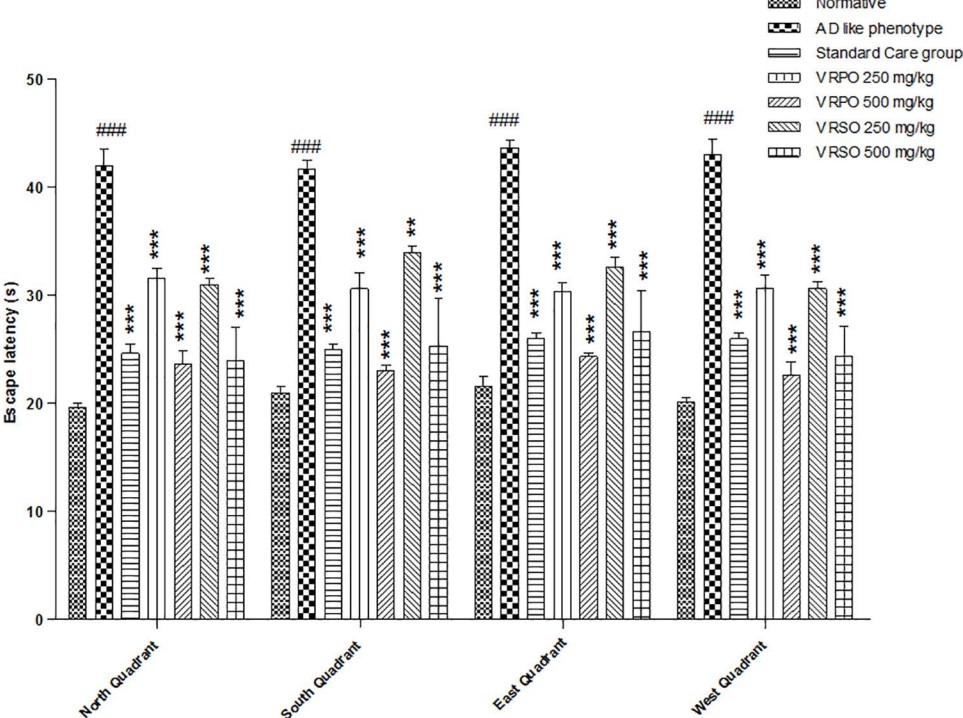

**Fig 5. Result of VRSO and VRPO on cognitive mapping and memory impairment in MWM task.** Statistical assessment (n = 10, in each group) Two-Way ANOVA showed significant variations compared to AD like phenotype group, with p-values indicating probabilities of * less than 5%, ** less than 1%, and *** less than 0.1% and compared to normative group ### less than 0.1%, respectively.

**Table 7. Outcome of VRSO and VRPO on movement, exploratory and anxiety like behavior in open field task.**

| Groups | Grooming | Rearing | Freezing moments | Number of lines crossed |
|---|---|---|---|---|
| | **Per 10 minute trial session** | | | |
| **Normative** | 5 ± 0.01 | 5 ± 0.02 | 0.00 | 34 ± 0.02 |
| **AD like phenotype** | 0### | 0.00### | 85 ± 0.01### | 11 ± 0.02### |
| **Standard Care Group** | 4 ± 0.03*** | 5 ± 0.1*** | 3 ± 0.02*** | 28 ± 0.02*** |
| **VRPO 250 mg/kg** | 2 ± 0.05* | 3 ± 0.05** | 5 ± 0.01*** | 24 ± 0.02*** |
| **VRPO 500 mg/kg** | 4 ± 0.01*** | 4.5 ± 0.03*** | 2 ± 0.05*** | 30 ± 0.02*** |
| **VRSO 250 mg/kg** | 3 ± 0.1** | 3.5 ± 0.2*** | 3 ± 0.06*** | 27 ± 0.02*** |
| **VRSO 500 mg/kg** | 5 ± 0.02*** | 5 ± 0.03*** | 0.00*** | 33 ± 0.02*** |

Statistical assessment (n = 10, in each group), Two-Way ANOVA showed significant variations compared to AD like phenotype group, with p-values indicating probabilities of

*less than 5%,

**less than 1%, and

***less than 0.1% and compared to normative group

###less than 0.1%, respectively.

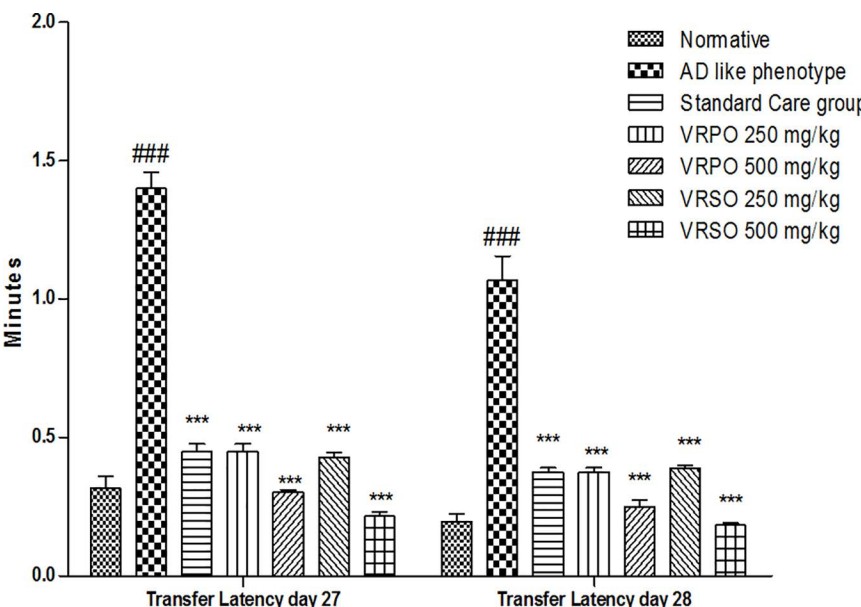

**Fig 6. Effect of VRPO and VRSO on exploratory and anxiety like behavior in elevated plus maze task.** Statistical assessment (n = 10, in each group), Two-Way ANOVA showed significant variations compared to AD like phenotype group, with p-values indicating probabilities of * less than 5%, ** less than 1%, and *** less than 0.1% and compared to normative group ### less than 0.1%, respectively.

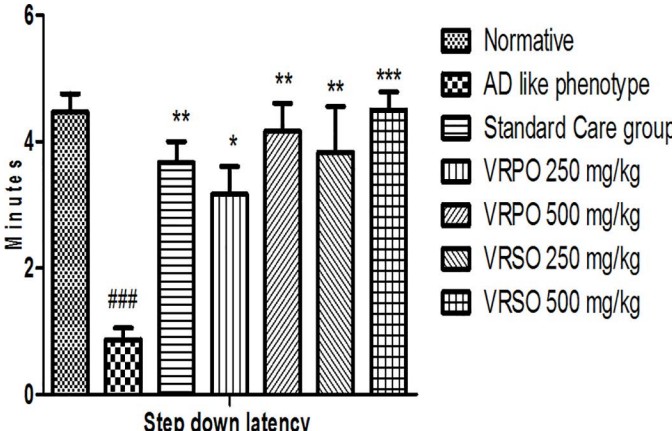

**Fig 7. Effect of VRPO and VRSO on fear conditioning and exploration in passive avoidance task.** Statistical assessment (n = 10, in each group), One-Way ANOVA showed significant variations compared to AD like phenotype group, with p-values indicating probabilities of * less than 5%, ** less than 1%, and *** less than 0.1% and compared to normative group ### less than 0.1%, respectively.

**Effect of VRPO and VRSO on hole board task.** It was manifested that AD like phenotype group showed a crucial decline (p < 0.05) in head dipping, locomotion, and exploratory way of behaving compared to normative group. In contrast, the standard care and VRPO and VRSO treatment groups exhibited significant improvements in exploratory behavior and head dipping, with a concentration related effect: the high concentration (500 mg/kg) showed the greatest improvement, followed by the low concentration (250 mg/kg) (Fig 8).

**Table 8. Effect of VRPO on spatial memory, learning and cognitive flexibility in Y-maze task.**

| Treatments | Total arm visits | Spatial memory Score in triadic response | Autonomic alternation Score (%) | Hemispheric dominance index |
|---|---|---|---|---|
| | Maze exploration metric | | | |
| Normative | 14 ± 0.11 | 4 ± 0.21 | 41 ± 0.31 | 0.16 ± 0.04 |
| AD like phenotype | 4.0 ± 0.1 | 1 ± 0.05 | 2 ± 0.02 | -0.4 ± 0.02 |
| Standard care | 10.0 ± 0.11 | 3 ± 0.04 | 38 ± 0.04 | 0.14 ± 0.03 |
| VRPO 250 mg/kg | 08 ± 0.12 | 3 ± 0.03 | 37 ± 0.05 | 0.11 ± 0.05 |
| VRPO 500 mg/kg | 13 ± 0.01 | 3.5 ± 0.02 | 39 ± 0.01 | 0.13 ± 0.01 |
| VRSO 250 mg/kg | 11 ± 0.04 | 2.5 ± 0.03 | 38 ± 0.03 | 0.12 ± 0.02 |
| VRSO 500 mg/kg | 14 ± 0.02 | 5 ± 0.01 | 41 ± 0.01 | 0.16 ± 0.04 |

Statistical assessment (n = 10, in each group), Two-Way ANOVA showed significant variations compared to AD like phenotype group, with p-values indicating probabilities of

* less than 5%,

**less than 1%, and

***less than 0.1% and compared to normative group

###less than 0.1%, respectively.

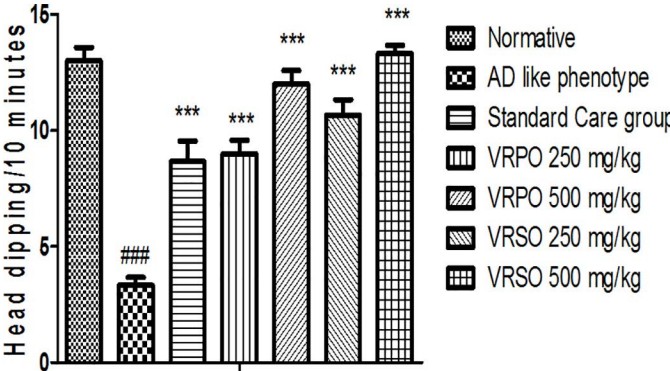

**Fig 8. Effect of VRPO and VRSO on fear conditioning and exploration on head dipping task.** Statistical assessment (n = 10, in each group), One-Way ANOVA showed significant variations compared to AD like phenotype group, with p-values indicating probabilities of * less than 5%, ** less than 1%, and *** less than 0.1% and compared to normative group ### less than 0.1%, respectively.

**Effect of VRPO and VRSO on wire hanging task.** Our study revealed a significant decrease (p < 0.05) in wire hanging time and a significant increase in falling down in AD like phenotype group compared to the normative group, standard care group, VRSO and VRPO treatment group. However, no significant differences (p > 0.05) were observed in suspending with wire and dropping times among the normative category, standard drug receiving category and VRSO & VRPO receiving category (Fig 9).

**Effect of VRPO and VRSO on first line antioxidant enzymes.** It was manifested that in AD like phenotype group the level of catalases, reduced glutathione and GPx remarkably lessened (p < 0.05) in comparison to normative group. However, due to perpetuation of oxidative stress brought on by D-galactose and $AlCl_3$ the level of lipid and proteins peroxidation increased, as manifested by significantly raised level of malonaldehyde in AD like phenotype group compared to normative group. VRPO and VRSO treatment markedly

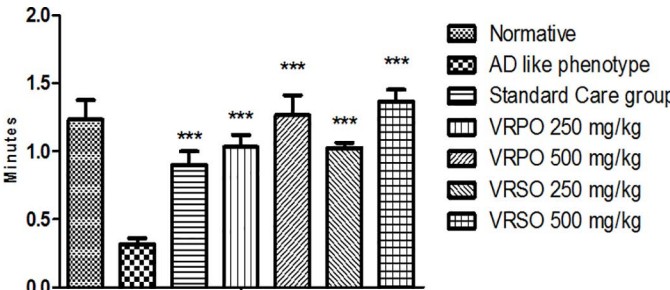

**Fig 9. Effect of VRPO and VRSO on wire hanging task.** Statistical assessment (n = 10, in each group), One-Way ANOVA showed significant variations compared to AD like phenotype group, with p-values indicating probabilities of * less than 5%, ** less than 1%, and *** less than 0.1% and compared to normative group ### less than 0.1%, respectively.

($p < 0.05$) recovered or mitigated the level of first line antioxidant enzymes dose dependently in treatment group. VRSO and VRPO treatment remarkably ($p < 0.05$) diminished the level of malonaldehyde (MDA) in VRSO & VRPO receiving groups similar to standard care and normative group (Table 9).

**Effect of VRPO and VRSO on acetyl cholinesterase.** It is revealed that d-galacatose and aluminum chloride insult treatment to AD like phenotype group induced depletion of acetylcholine via up-regulation in acetylcholinesterase activity. In AD like phenotype group the quantity of AChE crucially up surged ($p < 0.05$) resulting in decreased level of acetylcholine. However, in VRPO and VRSO treatment groups the level of acetyl cholinesterase decreased resulting in recovered level of acetylcholine and recovery of associated motor and non -motor functions congruent to normative and standard care treatment groups (Fig 10).

**Effect of VRPO and VRSO on serotonin, dopamine and nor-adrenaline.** In AD like phenotype group the level of principal neurotransmitters significantly declined compared to normative group. Whereas, standard care and VRSO and VRPO treatment significantly retained ($p < 0.05$) the concentration of these chemical messangers compared to AD like phenotype group (Table 10).

**Effect of VRPO and VRSO on messenger RNA expression of SFRP4, neurodegenerative and neuro-inflammatory molecular markers.** In AD like phenotype group the mRNA expression of Secreted Frizzled Related Protein 4 (SFRP4), interleukin 1 alpha (IL-1α), interleukin 1 beta (IL-1β), tumor necrosis factor alpha (TNF-α), acetyl cholinesterase (AChE), amyloid beta precursor protein (AβPP), beat secretase (β-secretase) significantly ($p < 0.05$) up-regulated compared to normal control category. However, VRPO & VRSO treatment significantly down-regulated the messenger ribonucleic acid (mRNA) expression of molecular markers of inflammation of nervous tissue and neuronal loss (Fig 11).

**Effect of VRPO and VRSO on protein expression of interleukin -6 (IL-6) and tumor necrosis factor alpha (TNF-α).** It was manifested by enzyme linked immune-sorbent assay (ELISA) that in AD like phenotype group the expression of inflammatory biomarkers remarkably raised ($p < 0.05$) contrast to normative group. However, VRPO and VRSO dose dependently decreased the expression of these inflammatory markers in treatment groups like normative and standard care group (Fig 12).

**Result of VRSO and VRPO on Serum SFRP4 level.** The serum concentration of SFRP4 was remarkably raised ($p < 0.001$) in AD like phenotypic category after administration of d-galactose and aluminum chloride. However, VRSO and VRPO dose dependently decreased the serum level of SFRP4 in treatment groups 500mg/kg>250 mg/kg (Fig 13).

**Table 9. Effect of VRPO and VRSO on first line antioxidant enzymes.**

| Groups | Superoxide dismutase IU/µL | Malonaldehyde (TBA mg/Ml) | Catalases IU/µL | Reduced glutathione(GSH) (µ/mg of protein) | Glutathione peroxidase(GPX) (µ/mg of protein) |
|---|---|---|---|---|---|
| Normative | 0.066 ± 0.1 | 67 ± 0.01 | 0.87 ± 0.2 | 0.78 ± 0.1 | 9.50 ± 0.2 |
| AD like phenotype | 0.023 ± 0.04### | 96 ± 0.03### | 0.44 ± 0.01### | 0.22 ± 0.02### | 3.20 ± 0.01### |
| Standard care | 0.052 ± 0.01*** | 75 ± 0.02*** | 0.72 ± 0.04*** | 0.65 ± 0.4*** | 7.7 ± 0.4*** |
| VRPO 250 mg/kg | 0.058 ± 0.03*** | 77 ± 0.01*** | 0.70 ± 0.01*** | 0.60 ± 0.32*** | 7.5 ± 0.02*** |
| VRPO 500 mg/kg | 0.061 ± 0.05*** | 70 ± 0.02*** | 0.79 ± 0.03*** | 0.70 ± 0.01*** | 8.4 ± 0.01*** |
| VRSO 250 mg/kg | 0.058 ± 0.04*** | 71 ± 0.01*** | 0.77 ± 0.07*** | 0.66 ± 0.02*** | 8.8 ± 0.2*** |
| VRSO 500 mg/kg | 0.062 ± 0.01*** | 72 ± 0.02*** | 0.83 ± 0.01*** | 0.72 ± 0.03*** | 9.2 ± 0.21*** |

Statistical assessment (n = 10, in each group), One-Way ANOVA showed significant variations compared to AD like phenotype group, with p-values indicating probabilities of

*less than 5%,

**less than 1%, and

***less than 0.1% and compared to normative group

###less than 0.1%, respectively.

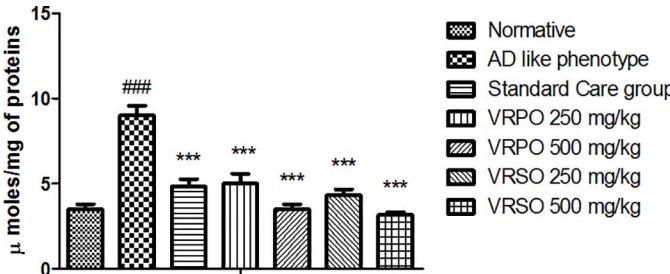

**Fig 10. Acetyl-cholinesterase inhibitory effect of VRPO and VRSO treatment.** Statistical assessment (n = 10, in each group), One-Way ANOVA showed significant variations compared to AD like phenotype group, with p-values indicating probabilities of * less than 5%, ** less than 1%, and *** less than 0.1% and compared to normative group ### less than 0.1%, respectively.

**Table 10. Effect of VRPO and VRSO on neurotransmitters.**

| Groups | Serotonin (µg/ mg of brain tissue) | Dopamine (µg/ mg of brain tissue) | Noradrenaline (µg/ mg of brain tissue) |
|---|---|---|---|
| Normative | 0.7 ± 0.021 | 0.37 ± 0.12 | 0.08 ± 0.01 |
| AD like Phenotype | 0.3### ± 0.11 | 0.12### ± 0.13 | 0.05### ± 0.04 |
| Standard care | 0.66*** ± 0.05 | 0.35*** ± 0.11 | 0.074*** ± 0.05 |
| VRPO 250 mg/kg | 0.59*** ± 0.03 | 0.22*** ± 0.03 | 0.065*** ± 0.011 |
| VRPO 500 mg/kg | 0.68*** ± 0.01 | 0.34*** ± 0.02 | 0.077*** ± 0.022 |
| VRSO 250 mg/kg | 0.57*** ± 0.04 | 0.21*** ± 0.04 | 0.64*** ± 0.12 |
| VRSO 500 mg/kg | 0.66*** ± 0.01 | 0.33*** ± 0.01 | 0.76*** ± 0.03 |

Statistical assessment (n = 10, in each group) One-Way ANOVA showed significant variations compared to AD like phenotype group, with p-values indicating probabilities of

*less than 5%,

**less than 1%, and

***less than 0.1% and compared to normative group

###less than 0.1%, respectively.

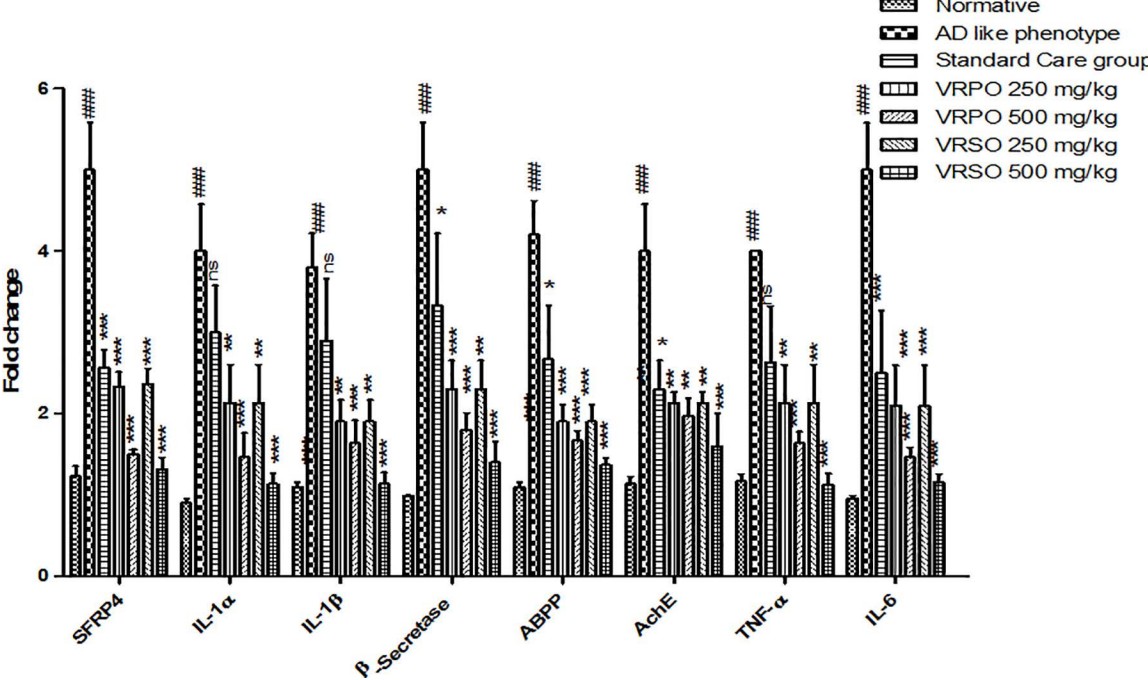

**Fig 11. Effect of VRPO and VRSO on mRNA expression of SFRP4, neurodegenerative and neuro-inflammatory biomarkers.**
Statistical assessment (n = 10, in each group), One-Way ANOVA showed significant variations compared to AD like phenotype group, with p-values indicating probabilities of * less than 5%, ** less than 1%, and *** less than 0.1% and compared to normative group ### less than 0.1%, respectively.

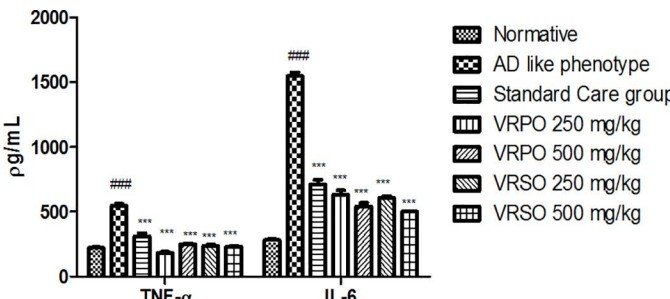

**Fig 12. Effect of VRPO and VRSO on protein expression of interleukin -6 (IL-6) and tumor necrosis factor alpha (TNF-α).** Statistical assessment (n = 10, in each group), One-Way ANOVA showed significant variations compared to AD like phenotype group, with p-values indicating probabilities of * less than 5%, ** less than 1%, and *** less than 0.1% and compared to normative group ### less than 0.1%, respectively.

**Effect of VRPO and VRSO on histopathology of brain tissues.** It was manifested through histo-pathological analysis that treatment with VRPO and VRSO enriched extracts mitigated neurodegenerative hallmarks in brain tissues. D-galactose and aluminum chloride treatment induced the genesis of hyperphosphorylated tau proteins clumps inside neurons, pigmentation and plaques. The neuronal cell count also decreased in AD like phenotype group. However, normative, standard care and VRPO and VRSO treated group showed normal architecture of brain tissues (Fig 14). Fig 15 showed the scoring or grading system of histo-pathological interpretations. It was manifested that highest scores of neurofibrillary

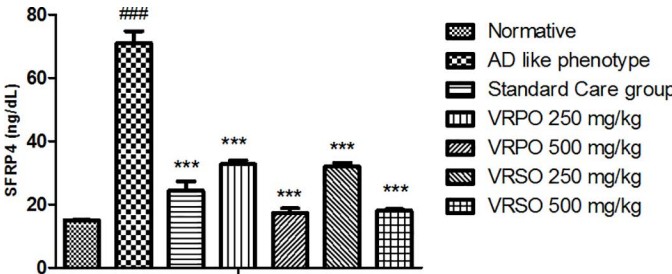

**Fig 13. Effect of VRPO and VRSO on serum level of SFRP4.** Statistical assessment (n = 10, in each group), One-Way ANOVA showed significant variations compared to AD like phenotype group, with p-values indicating probabilities of * less than 5%, ** less than 1%, and *** less than 0.1% and compared to normative group ### less than 0.1%, respectively.

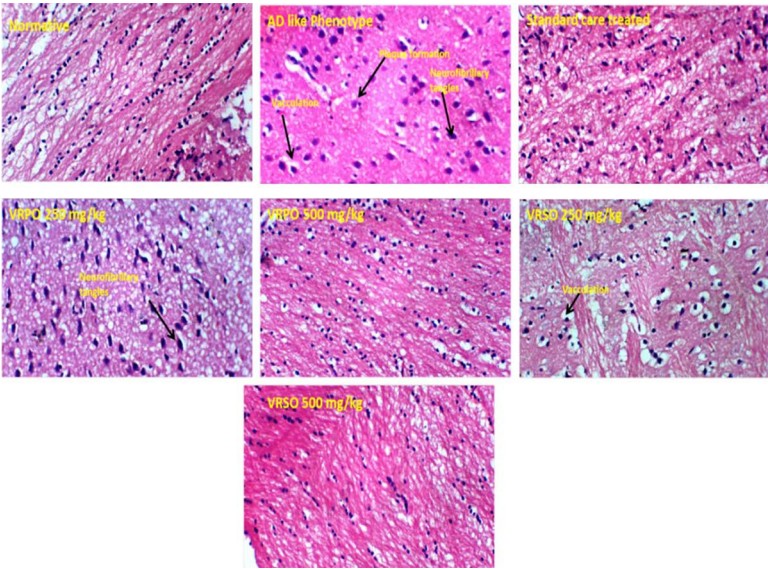

**Fig 14. Effect of VRPO and VRSO on histopathology of brain tissues after hematoxylin and eosin staining.**

tangles, senile plaques, neuro-inflammation and neuronal loss were marked to AD like phenotype group compared to standard care, VRSO and VRPO treatment groups. It was evident that VRSO and VRPO treatment at higher doses 500 mg/kg significantly recovered (p < 0.01) the neuronal architecture compared to AD like phenotype group.

## Discussion

This study explored the therapeutic potential of *V. radiata* (green beans) also known as mung beans, extract in pumpkin oil and soya bean oil in an AlCl$_3$ and D-galactose-generated Alzheimer's disease (AD) rat model. Our findings demonstrate that these natural extracts significantly improved cognitive function, reduced oxidative stress, and inhibited amyloid-β aggregation in the brain, as evidenced by behavioral studies, biochemical analysis, histopathology, and RT-PCR analysis.

Neurodegenerative diseases like Parkinson's and Alzheimer's share common features, including protein accumulation and oxidative stress, but current treatments only address

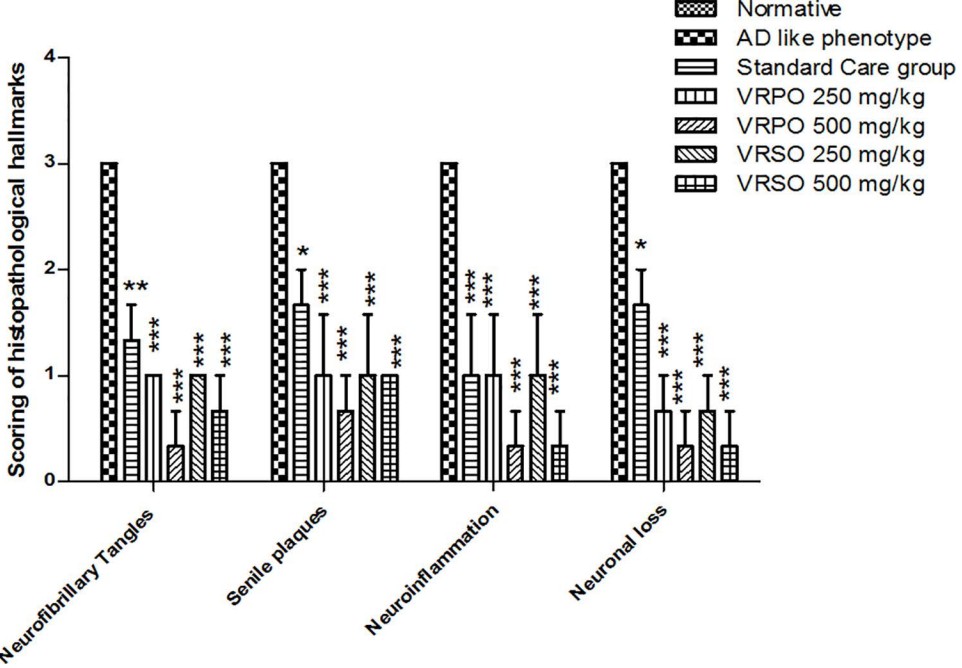

**Fig 15. Scoring of histo-pathological findings of AD associated hallmarks.** (Neurofibrillary tangles 0 (none), 1(Sparse), 2(Moderate), 3(Severe)), (Senile plaques 0 (none), 1 (Sparse), 2 (Moderate), 3 (Severe)), Neuro-inflammation 0 (none), 1(Mild), 2(Moderate), 3(Severe)), Neuronal loss 0 (none), 1(Mild), 2(Moderate), 3(Severe)). Statistical assessment One-Way ANOVA showed significant variations compared to AD like phenotype group, with p-values indicating probabilities of * less than 5%, ** less than 1%, and *** less than 0.1% and compared to normative group ### less than 0.1%, respectively.

symptoms, not causes. Nutrition, particularly lipids like vegetable and animal oils or fatty acids, may help prevent or slow disease progression when incorporated into diet. Modern research suggests these lipids can inhibit cytotoxicity and oxidative stress, and future thera-peutic/preventive approaches may involve improve delivery methods, such as lipid encapsula-tion, to target the brain and other vital organs.

GC-MS study declared the existence of remarkable bioactive elements in VRSO and VRPO extract. In VRSO Linoleic acid, 9-Octadecenoic acid, 4-Phenylpyridine, gamma.-Tocopherol, Vitamin E, Stigmasterol, Gamma-Sitosterol and Chrysin were identified through NIST libraray. Vitamin E is a fat-soluble vitamin with antioxidant properties, gamma-tocopherol is part of the vitamin E classification group found in corn and soybean oils [23,31]. Stigmasterol and gamma-Sitosterol are plant sterol with neuroprotective effects [31]. Chrysin is a flavonoid with neurodefensive potential [30]. Consistent to our findings a study reported that mung bean or green gram, an abundant resource of neuro-protective polyphenols, may be a potential dietary intervention for Alzheimer's disease (AD). Quanti-tative analysis identified nineteen phenolic compounds, including phenolic acids and flavo-noids, with vitexin, isovitexin, sinapic acid, and ferulic acid speculated to be key biologically active compounds. These compounds likely exert neuroprotective effects through mech-anisms including suppression of synthesis of β-amyloido proteins, tau hyperphosphory-lation, free radical damage, and neuroinflammation, as well as encouragement of removal of unnecessary cellular componemts and AchE enzyme action. Principally, germination altered the antiproliferative and neuroprotective phenols level, with changes in individual compound levels [8].

Alzheimer's disease (AD) is a common form of memory loss or impairment in normal brain functioning with restricted treatment options. Natural ingredients, such as *Vigna radiata* extract in pumpkin and soya bean oil have gained attention for its potential in treating neurological disorders, including AD. A previous study investigated the *in vitro* anticholinesterase and neuronal function promoting action of these plants. The extract of *V. radiata* prepared in ethyl acetate ($C_4H_8O_2$) displayed significant acetylcholine esterase (AChE) inhibitory action, with $IC_{50}$ values of 286.40 µg/mL. Additionally, extract exhibited dose-dependent neuroprotective effects against β-amyloid-generated cellular toxicity in SHSY-5Y neuroblastoma cells. The results proposed that *V. radiata* may prove a promising candidate for the synthesis of anti-Alzheimer's medications [32].

Therefore, keeping in view the *in vitro* neuroprotective potential of *V. radiate*, current study revealed cerebro-protective ability of *V. radiate* vegetable oil extracts in aluminum chloride and d-galactose induced AD like animal model. Behavioral studies revealed that VRSO and VRPO treated rats exhibited improved memory and learning abilities, indicating a potential reversal of AD-related cognitive decline. In behavioral studies during Morris water maze task, Y-maze task, elevated plus maze task, open field task, wire hanging and hole board test it is revealed that VRSO and VRPO significantly retained spatial learning, memory, cognitive flexibility, exploratory behavior, spontaneous alteration, muscular endurance and motor co-ordination.

Similar to our findings green moong bean (*Vigna radiata*) sprouts were investigated for their anti-Alzheimer potential using behavioral models in Swiss mice. Mice were divided into 23 groups and administered 2, 4, or 8% w/w moong bean sprouts (MBS) in their diet for 15 days [33]. Results showed that MBS significantly improved immediate memory and episodic memory in the Elevated Plus Maze and Passive Avoidance Paradigm models, respectively. MBS also reversed ethanol- and diazepam-induced memory deficits. Additionally, MBS increased brain glutathione levels, decreased acetylcholinesterase activity, and reduced malondialdehyde levels, suggesting a mechanism of action involving acetylcholinesterase inhibition, antioxidant activity, and neuroprotection via phytoestrogens. Overall, the outcomes declared that *V. radiata* sprouts possess promising anti-Alzheimer potential [34].

Biochemical tests indicated a crucial reduction in oxidative stress markers, such as malondialdehyde, and raise in the level of enzymes responsible for scavenging free radicals, like superoxide dismutase and catalase, in the brain tissues of VRSO and VRPO treated rats.

Similar to our results a study investigated the defensing effects of Ginkgo biloba extract (EGb) and pumpkin seed oil (PSO) against neouronal toxicity produced by rotenone in rodents. Both EGb and PSO improved dopamine and norepinephrine levels, antioxidant capacity, and lipid peroxidation, with EGb showing a more pronounced protective effect. The study suggests that EGb may prove helpful against neuronal damage caused by atmospheric neurotoxic chemicals, and PSO may have a role in limiting oxidative stress [35].

Acetylcholinesterase (AChE) plays a crucial role in nerve function and signal transmission, but excessive activity during aging can deplete acetylcholine (ACh), leading to impaired neural communication and serious illnesses like dementia and Alzheimer's Disease (AD) [36]. Inhibiting AChE activity to increase brain ACh levels is a proven approach for managing AD.

In current work, VRSO and VRPO decrease the excessive activity of this enzyme and restored the level of acetylcholine. Our study congruent to previous work cited the recent advances in using extracts and compounds obtained natural resources as AChE regulatory substances [37]. These include polyphenol-rich extracts, proteins, peptides, terpenoids, and carotenoids, which have shown efficacy in reducing brain AChE levels and improving memory functions [38].

In AD like phenotype model animals the significant decline in neurotransmitters like adrenaline, acetylcholine, dopamine and serotonin was observed. However, upon treatment with VRSO and VRPO level of these transmitters was markedly improved consistent to previous findings [8]. In previous work it was reported that presence of bioactive compounds such as Vitexin, Isovitexin, Sinapic acid, Ferulic acid *V. radiate* modulated acetyl-cholinesterase activity and level of acetylcholine [8].

VRSO and VRSPO downregulated the mRNA expression of neuronal deteriorating and neuro-inflammatory markers in RT-PCR analysis. AD a complex neurodegenerative disorder its pathogenesis involved the chronic neuro-inflammation, deposition of amyloid-β plaques intiated by central cascades of proinflammatory cytokines [39,40]. IL-1α, IL-1β, TNF-α, and IL-6 play a critical role in AD associated neuroinflammation. Acetylcholinestersae [41], Beta-secretase [42] and amyloid-β precursor protein (ABPP) [43] are key enzymes mainly involved in pathology of AD and production of amyloid-β deposits, while SFRP4 as modulator of Wnt/β-catenin signaling pathway also implicated in AD pathology [44]. Current findings revealed that VRSO and VRPO treatment markedly decreased the mRNA expression of IL-1α and IL-1β and therefore decreased the neuroinflammation and accumulation of β-amyloids significantly in treated groups compared to AD like phenotype group. It is manifested that both VRSO and VRPO markedly down-regulated the expression of TNF-α and IL-6 which are involved in immune response and neuroinflammation and subsequent AD pathogenesis. VRSO and VRPO markedly downregulated the mRNA expression of AD associated genes β-Secretase and ABPP, resulting in reduction in β-amyloid accumulation, senile plaque formation and formation of neurofibrillary tangles. Wnt/β-catenin signaling play a crucial role in synaptic plasticity, neurogenesis, neuronal integrity and overall brain health [45]. Oxidative stress or other neurotoxic substances down-regulated this signaling and induce AD like symptoms. SFRP4 (Secreted Frizzled Related Protein 4) is a down-regulator or negative regulator of this signaling. VRSO and VRPO treatment dosedependently restored Wnt/β-catenin signaling through downregulation in mRNA expression of SFRP4 suggesting a novel approach to treat AD.

Similarly, a study investigated that during menopause, decreased estrogen production can lead to reduced brain metabolism and increased risk of neurodegeneration. This study investigated the potential neuroprotective effects of pumpkin seed oil nanoparticles (PSO-NE) in an experimental postmenopausal model. The results showed that PSO-NE significantly increased estrogen levels, reduced neuro-inflammatory markers, and improved brain health, suggesting a prophylactic effect on neuro-inflammatory interactions [46].

In VRSO, soybean isoflavone (SIF) is a polyphenol with strong antioxidant activity, with genistein being the major isoflavone in soy foods. Research suggests that SIF may help alleviate deterioration of neuronal cells in various diseases like AD, and appropriate intake may diminish the threat of being suffered from AD. This research work analyze various aspects of AD pathogenesis and its association with SIF consumption, providing insights for AD prevention [47].

Similar to our work on vegetable oils used in extraction procedure for preparation of neuroprotective remedial agents. A study investigated the protective effects of three vegetable oils (olive, corn, and perilla) against intellectual disability in an AD in rats. Perilla oil, rich in alpha-linolenic acid (ALA), showed the most significant attenuation of cognitive impairment and reduced oxidative stress, inflammation, and improved brain function. The findings suggest that ALA-rich perilla oil may have potential for preventing or treating neurodegenerative diseases like AD [48].

VRSO have soy isoflavones (SI) have been suggested to have neuroprotective effects against AD, but their mechanisms of action are not well understood. This study found that SI

administration improved cognitive performance and enhanced cholinergic system function in mental disability brought on by scopolamine in rodent model. SI also suppressed oxidative stress and upregulated key signaling pathways, including ERK, CREB, and BDNF, in the hippocampus, suggesting its potential in recovering neuronal damage in many diseases like AD [49,50].

Histopathological examination revealed decreased level plaques and hyperphosphory-lated tau protein aggregates in the hippocampus and cortical region of standard and extract receiving animals, suggesting a possible neuroprotective effect [51]. RT-PCR analysis further confirmed the downregulation of genes involved in inflammation and oxidative stress, such as TNF-α and IL-1β. The outcomes revealed that *V. radiata*, or mung beans pumpkin and soya bean oil extracts may be potential natural remedies for AD treatment, targeting multiple pathological mechanisms. The neuroprotective properties of these extracts may be credited to their reducing inflammation antioxidant, and neurotrophic actions.

Our study highlights the importance of exploring natural products as alternative or complimentary therapies for AD, particularly in light of the limited success of current treatments. However, more study is required to clarify the primary mechanisms and raising the medicinal activities of these extracts. The findings of this research work offer a promising base for upcoming studies to utilize *V. radiata*, pumpkin oil and soya bean oil extract in AD prevention and treatment.

## Conclusion

This study demonstrates the therapeutic potential of *V. radiata* extract in pumpkin oil and soya bean oil in an AlCl$_3$ and D-galactose generated Alzheimer's disease (AD) rat model. The findings suggest that these natural extracts significantly improved cognitive function, reduced oxidative stress, and inhibited amyloid-β aggregation in the brain. The neuro-protective potential of VRSO & VRPO was probably be due to extracts' free radical trapping, anti-inflammatory, and neurotrophic actions. The study highlights the importance of exploring natural products as alternative or complementary therapies for AD, particularly in light of the limited success of current treatments. Though, additional studies are necessary to explain the principal mechanisms and boost the medicinal potential of these extracts. The outcomes of this research work provide an ensuring foundation for future researches using *V. radiate* or mung beans pumpkin oil, and soya bean oil extract in AD prevention and treatment. The neuroprotective effects of these extracts may offer a potential natural remedy for AD, targeting multiple pathological mechanisms. Generally, this research study adds to the accelerating pool of research on the neuroprotective activities of natural products and highlights the potential of *V. radiata*, pumpkin oil, and soya bean oil extract as a natural therapeutic agent for AD treatment.

## Supporting information

**S1 File.** **(1) Result of VRSO and VRPO on cognitive mapping and memory impairment in MWM task Mean values to generate graph. (2) Mean values to generate graph Effect of VRPO and VRSO on exploratory and anxiety like behavior in elevated plus maze task. (3) Effect of VRPO and VRSO on fear conditioning and exploration in passive avoidance task. Mean values to generate graph. (4) Effect of VRPO and VRSO on fear conditioning and exploration on head dipping task. Mean values to generate graph. (5) Effect of VRPO and VRSO on wire hanging task. Mean values to generate graph. (6) Acetyl-cholinesterase inhibitory effect of VRPO and VRSO treatment, Mean values to generate graph. (7) Effect of VRPO and VRSO on mRNA expression of SFRP4, neurodegenerative and neuro-inflammatory biomarkers Mean values to generate graph.**
(DOCX)

## Author contributions

**Conceptualization:** Haroon Amin, Shazia Anwer Bukhari, Zunera Chauhdary, Naheed Akhter, Maria Saleem.

**Data curation:** Haroon Amin, Shazia Anwer Bukhari, Zunera Chauhdary, Naheed Akhter, Maria Saleem.

**Formal analysis:** Haroon Amin, Shazia Anwer Bukhari, Zunera Chauhdary, Naheed Akhter, Maria Saleem.

**Investigation:** Haroon Amin, Zunera Chauhdary, Naheed Akhter.

**Methodology:** Haroon Amin, Zunera Chauhdary.

**Project administration:** Zunera Chauhdary.

**Resources:** Shazia Anwer Bukhari.

**Software:** Shazia Anwer Bukhari.

**Supervision:** Haroon Amin, Shazia Anwer Bukhari, Zunera Chauhdary, Naheed Akhter.

**Validation:** Shazia Anwer Bukhari.

**Visualization:** Maria Saleem.

**Writing – original draft:** Haroon Amin, Zunera Chauhdary, Naheed Akhter, Maria Saleem.

**Writing – review & editing:** Haroon Amin, Shazia Anwer Bukhari, Zunera Chauhdary, Naheed Akhter, Maria Saleem.

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
