## [Decision Letter · Decision Letter 0]

20 Dec 2024

PONE-D-24-45075Vigna radiata Extracts in Pumpkin and Soya Bean Oil: A Novel Therapeutic Approach for Alzheimer's diseasePLOS ONE

Dear Dr. Bukhari,

Thank you for submitting your manuscript to PLOS ONE. After careful consideration, we feel that it has merit but does not fully meet PLOS ONE’s publication criteria as it currently stands. Therefore, we invite you to submit a revised version of the manuscript that addresses the points raised during the review process.

We look forward to receiving your revised manuscript.

Kind regards,

Vara Prasad Saka

Academic Editor

PLOS ONE

Journal Requirements:

Reviewers' comments:

Reviewer's Responses to Questions

**Comments to the Author**

1. Is the manuscript technically sound, and do the data support the conclusions?

Reviewer #1: No

Reviewer #2: Partly

Reviewer #3: Partly

2. Has the statistical analysis been performed appropriately and rigorously? 

Reviewer #1: No

Reviewer #2: No

Reviewer #3: Yes

3. Have the authors made all data underlying the findings in their manuscript fully available?

Reviewer #1: No

Reviewer #2: No

Reviewer #3: Yes

4. Is the manuscript presented in an intelligible fashion and written in standard English?

Reviewer #1: No

Reviewer #2: No

Reviewer #3: Yes

5. Review Comments to the Author

Reviewer #1: This is a very ambitious report of the components and effects of plant extracts on an animal model of Alzheimer's disease. The authors attempt to chemically characterize the extract, perform toxicology, behavioral, brain biochemistry and histological studies all in one manuscript leading to the omission of critical design elements.

Many needed descriptions of the experimental design are absent. Rats and mice are indicated (Materials and Methods section under Animals) as being used but it is not always clear which experiments used rats and which used mice. The number and species of experimental units in all data figures and tables is unspecified. The timing, frequency and duration of experimental therapeutic dosing is not specified. Was some criteria for verification of AD phenotype used, what criteria were used for inclusion or exclusion of animals? Were evaluators blinded as to the group? Animals randomized?

Figures are not interpretable as given. Figure legends do not include statistical tests utilized, the meaning of multiple asterisks, the number of experimental units or the species. Authors need to specify the exact number of experimental units allocated to each group and the total number in each group for each figure. In addition, if groups are smaller than 10 it is preferable to plot each animals data point, if groups are greater than 10 then SD (or median and range) is preferable to SEM. ANOVA with repeated measures is specified but what measures were repeated is not mentioned (Figures 4 through 9, are those measures before and after treatment). Were multiple therapies applied in the same animal? Trinaing is mentioned for behavioral tasks but time frame and criteria for adequacy not mentioned

Authors indicate that all relevant data are included in the manuscript but this is not accurate as only means and SEMs (with no number of experimental units) are included. PLOS ONE Policy states "PLOS journals require authors to make all data necessary to replicate their study’s findings publicly available without restriction at the time of publication. .... For example, authors should submit the following data:

• The values behind the means, standard deviations and other measures reported;

• The values used to build graphs;

• The points extracted from images for analysis. "

Figure 9 is mislabelled.

Figure 13 suggest that a (single?) dose of extract restores AD-induced brain architecture. It seems unlikely that these are representative images.

Many references are incomplete.

Authors may benefit by consultingthe ARRIVE checklists for pre-clinical animal studies - https://www.equator-network.org/reporting-guidelines/improving-bioscience-research-reporting-the-arrive-guidelines-for-reporting-animal-research/

Reviewer #2: Thank you for your manuscript. I explain my answers to the question#1-4. #1.I found that your conclusion is mostly supported by the data, but I couldn't find the data that the content of acetylcholine decreased as you mentioned in the manuscript. In addition, I don't think that the high concentration (500 mg/kg) showed the greatest improvement, followed by the low concentration(250 mg/kg) (Figure 7) as you mentioned because there was no statistical analysis. #2. As I mentioned in #1, because of the lack of stastical analysis in some data, I would like you to confirm the statistics. #3. As I mentioned in #1, I couldn't find the data that the content of acetylcholine decreased as you mentioned in the manuscript. And it seemed unclear that you described the recovery of associated motor and non motor functions in Effect of VRPO and VRSO on acetyl cholinesterase (Figure 9). #4. I thought the majority of the manuscript was good, but I found some of it to be overly worded, e.g. "curing" (there is no definition of "curing" and it was only the animal model of AD that is being evaluated. I thought that it was better to use "administraion" instead of curing). <additional comments=""> You need to provide more information about anethesia in page 4 and material and methods. In material and methods, it is necessary to clarify which sex, how many animals, and what age in weeks were used in each experiment, and to specify that the test substance treatment group is hitting the AD-like model (readers could guess the situation but it is needed to be written). In addition, the route of administration of the test substance, duration of administration, and the specified timing at which each evaluation was performed, are also needed in material and methods. In results, under the title of each figure or table, a short description of the data, the number of animals, the meaning of error bars and statistical symbols should be written. And could you confirm "Figure 1: GC-MS spectrum of detected bioactive compounds in VRPO extract”? That is the same as Figure 2.

I guess Fig 1 represents the data of VRSO, not VRPO. In the title of "Figure 4: Results of VRPO on cognitive mapping and memory impairment in the MWM task” and “Table 7: Results of VRPO on motor, exploratory, and anxiety-like behaviors in open field tasks”, VRSO seems to be needed in the title as well. The title "Figure 9: Effectiveness of VRPO and VRSO in wire hanging task" and the data in Figure 9 seems not to match. In Table 10: Effect of VRPO and VRSO on neurotransmitters, "ug/ug" looks like too much. Was there any change in the condition of the base animals? In Figure 11: Effects of VRPO and VRSO on protein expression of interleukin-6 (IL-6) and tumor necrosis factor-alpha (TNF-alpha), the content of cytokine seems like too high for the normative group. Lastly, there are some sentences that are missing citations and typographical errors, please include and correct them. Thank you for your cooperation.</additional>

Reviewer #3: Title: “Vigna radiata Extracts in Pumpkin and Soya Bean Oil: A Novel Therapeutic Approach for Alzheimer's disease”

This study investigates the neuroprotective effect of V. radiate pumpkin and soya bean oil extract in motor dysfunction and mental cognitive decline in Alzheimer’s disease (AD) rat model.

The subject is interesting and address important issue. The research design is sound, well conducted, and the manuscript may be suitable for publication after major modifications.

Some points of concerns are listed below:

Introduction:

-Some abbreviations are mentioned without their full name such as NFTs, no list of abbreviations is provided.

Material and methods:

-It is stated that “Animals of either sex, mice (25-30 g) or rats (150-200 g), were taken…”.

Which type of animals used in study & what number in each group?

-It is stated that “AD was induced in experimental animals through oral administration of aluminum chloride and d-galactose at a dose of 150 mg/kg each for 28 days”.

Is this dose is given to animals daily or once only during this study period?

-The dosage of drug treatment to various experimental groups is not clear.

Are the drugs given every day or once during study? Is this treatment before, with or after induction of AD?

-A lot of neurobehavioral tests were used in the study without specific rational or relation to the AD symptoms.

-In Biochemical analysis of oxidative stress molecular markers and for chemical transmitters of brain, Which area of brain is used or whole brain homogenate is used?

-In Histopathological analysis, which area of brain is processed and used for histopathological examination?

-Also, in PCR amplification, the gene expressions are tested in which area of the brain?

-In ELISA test, the quantities of proinflammatory cytokines were assessed in serum or in brain tissue?

Results:

-It is stated that “It was manifested that in AD like phenotype group the level of catalases, malonaldehyde, reduced glutathione and GPx remarkably lessened”

The level of malondialdehyde is elevated, this need to be revised and corrected as shown in table 9.

-It is stated that “Effect of VRPO and VRSO on acetyl cholinesterase” shown in (Figure 9). But, Figure 9: Effect of VRPO and VRSO on wire hanging task

-The Effect of VRPO and VRSO on histopathology of brain tissues is assessed by subjective observation without any grading or scoring system for pathological lesions.

Discussion and conclusion:

The discussion includes many broad and vague terms with generalization, instead of concentrating on specific interpretation of the results. There is a lot of redundancy and repetition in many parts with overshooting conclusions.

-Some sentences are confusing and difficult to understand for example: “VRSO and VRSPO downregulated the mRNA expression of neuronal deteriorating and neuroinflammatory markers in RT-PCR analysis. Please revise and clarify.

-The whole manuscript should be revised for some typing and grammar corrections.

References:

Some references are deficient with missing data such as references 5, 16, 19.

Please revise.

6. PLOS authors have the option to publish the peer review history of their article (what does this mean? ). If published, this will include your full peer review and any attached files.

**Do you want your identity to be public for this peer review?** For information about this choice, including consent withdrawal, please see our Privacy Policy .

Reviewer #1: No

Reviewer #2: No

Reviewer #3: No

---

## [Author Response · Author response to Decision Letter 1]

8 Feb 2025

Dear reviwers thanks for very good and valueable suggestions, i adress all comments positively, please consider them a humble request. thanks

please process it fast as required for Student PhD notification please

---

## [Decision Letter · Decision Letter 1]

3 Mar 2025

Vigna radiata Extracts in Pumpkin and Soya Bean Oil: A Novel Therapeutic Approach for Alzheimer's disease

PONE-D-24-45075R1

Dear Dr. Bukhari,

We’re pleased to inform you that your manuscript has been judged scientifically suitable for publication and will be formally accepted for publication once it meets all outstanding technical requirements.

Kind regards,

Vara Prasad Saka

Academic Editor

PLOS ONE

Additional Editor Comments (optional):

Reviewers' comments:

Reviewer's Responses to Questions

**Comments to the Author**

1. If the authors have adequately addressed your comments raised in a previous round of review and you feel that this manuscript is now acceptable for publication, you may indicate that here to bypass the “Comments to the Author” section, enter your conflict of interest statement in the “Confidential to Editor” section, and submit your "Accept" recommendation.

Reviewer #3: All comments have been addressed

2. Is the manuscript technically sound, and do the data support the conclusions?

Reviewer #3: Yes

3. Has the statistical analysis been performed appropriately and rigorously? 

Reviewer #3: Yes

4. Have the authors made all data underlying the findings in their manuscript fully available?

Reviewer #3: Yes

5. Is the manuscript presented in an intelligible fashion and written in standard English?

Reviewer #3: Yes

6. Review Comments to the Author

Reviewer #3: Thanks to the authors for addressing the comments of reviewers in the revised manuscript that became suitable for publication

7. PLOS authors have the option to publish the peer review history of their article (what does this mean? ). If published, this will include your full peer review and any attached files.

**Do you want your identity to be public for this peer review?** For information about this choice, including consent withdrawal, please see our Privacy Policy .

Reviewer #3: No

---

## [Editor Report · Acceptance letter]

PONE-D-24-45075R1

PLOS ONE

Dear Dr. Bukhari,

I'm pleased to inform you that your manuscript has been deemed suitable for publication in PLOS ONE. Congratulations! Your manuscript is now being handed over to our production team.

Kind regards,

on behalf of

Dr. Vara Prasad Saka

Academic Editor

PLOS ONE